Published in FAST Workshop on Smalltalk Related Technologies (11/2022)

# Live Proof-by-Induction

**Boris Shingarov**                                                      *shingarov@labware.com*
*LabWare*

**Reviewed on OpenReview:** *https://openreview.net/forum?id=AaMvINlc7d*

## Abstract

Refinement types are a powerful formalism with many applications in program verification, compiler optimization, etc. We present our work-in-progress on MachineArithmetic, an implementation of refinement types in Smalltalk. Our current results are empirical, successfully running tests in the LiquidHaskell benchmark suite.

> They say you do not understand a topic in mathematics until you can teach it. And you do not really understand it until you can prove it in Coq. And you do not really *really* understand it until you can prove it in Agda.
>
> — Russell O'Connor

> And you do not really *really REALLY* understand it until you have debugged why the proof fails.
>
> — this author

## 1  Motivation

Designing an optimized Smalltalk Virtual Machine for direct execution on modern, post-Moore's-law processors presents new classes of challenges which processors with traditional sequentially-consistent ISAs did not pose. The lack of sequential consistency in the memory model not only breaks the assumptions of the common compiler optimizations (Morisset, 2017); the very approach where the systems programmer writes and tests code based on some intuitive argument why the code works, is — out of necessity — being replaced by programmatic manipulation of said argument (suitably formalized). In particular, when it comes to ISA semantics, we shift from relying exclusively on our informal understanding of the processor (which we gained by reading the processor manual, experimenting, falling into pit-traps etc.) towards machine-assisted analysis of formal specifications, e.g. using tools such as Sail (Armstrong et al., 2019; Wassell et al., 2018).

As Smalltalkers, we would of course prefer to have these tools homogeneous with our Smalltalk system therefore enjoying all the benefits of the Smalltalk paradigm (and even more so if we want to write the VM in Smalltalk). For this ambition, at a minimum we need to have a suitable automated-reasoning engine. This author's efforts in building such engine and such semantic tools, have led him to realize that in Smalltalk-80 our basic constructions lack certain desirable mathematical properties. In our arsenal of concepts, there is no structure that would satisfy the axioms of a topos; therefore we can't build a machine logic. Our classes do not compose; as well, they can be prone to expressing details of representation as opposed to intended meaning.

Let's consider a simple example.

LiveTyping (Wilkinson, 2019) aids the understanding of program behavior by collecting type information as the program runs. The nature of this information, is class membership. If I call the method

```
fourtyTwo
    ^42
```

a thousand times, from ten different call sites, LiveTyping will tell me that the return value was always a `SmallInteger`. How suitable is the granularity of this typing to my understanding of the program? is it too fine? too coarse?

Consider `Number>>abs`. If my program run contains the send: `42 negated abs` and the send: `42 factorial abs`, LiveTyping will report about `SmallInteger` and `LargePositiveInteger`. Here the question of granularity-relevance becomes especially evident (should `42` be a `SmallInteger`? a `Number`? a `PositiveInteger`?)[1] and the problem is not in LiveTyping but in the classes themselves: the fact that `42 abs` is a `SmallInteger`, is telling me something about how integer arithmetic is implemented on top of the underlying hardware, but not much about the meaning of `#abs`.

When I am reasoning about what `#abs` is supposed to return, at a minimum I expect some mention along the lines of `NonNegativeInteger` (to capture the "positivity" aspect) — something we do not have. Even if we did, we still can't compose it, for example, by intersecting with `SmallInteger` to obtain `SmallNonNegativeInteger` (standing for the integers that are both *small* and *nonnegative*): classes don't afford us logical connectives between propositions.

As another datapoint to make the emerging pattern more visible, let's consider the (nondestructive) function "add 1":

```
add1
    ^self + 1
```

If I throw a million different SmallIntegers at this function, I may (statistically very probably) get back a million SmallIntegers, leaving me with the impression that add1 **always** returns a SmallInteger. Such claim is both an underapproximation and an overapproximation. On the one hand, it is too weak to explain why we would reject

```
add1
    ^42
```

as an implementation for add1; on the other hand, it is too strong to be correct (i.e. SmallInteger not a superset of all possible values) becaues it misses the edge case where adding 1 overflows into LargeInteger. Thus our class-hierarchy of numbers starts to feel like the wrong level of abstraction to reason about what it means "to add 1 to an integer."

As a final illustrative example let's consider `Collection>>size`. If I have four tests, sending `#size` to `'big'`, `'pig'`, `'can'`, `'dig'`, in all cases the returned value is the same object `3`. What granularity would be appropriate to capture the essence of `#size`? Is it `SmallInteger`? `Number`? the particular object 3? [2]

One could, of course, attempt to construct a class such as PositiveInteger, EvenInteger, etc. This immediately runs into some questions. How should these classes connect to other classes in the Number hierarchy? How do they connect to each other? What does "class membership" now mean? Is `42` an instance of PositiveInteger? EvenInteger? SmallInteger?

---

[1] In a real-life debugging session, the complexity can easily derail the programmer's thinking so much they start asking questions as confused as "is the SmallInteger not PositiveInteger because of the negated?"

[2] Another dimension is that, size is polymorphic not only in its receiver, but also the receiver's elements; below we shall see why this is a major problem, and what we propose to do about it.

By all this, I don't mean to say that classes are broken or defective. The above few paragraphs try to delineate the task of creating a higher-level framework — which ideally would be embedded in our existing Smalltalk language as naturally as possible — suitable for building a computer logic in Smalltalk, which, in turn, would be suitable as a basis for automated reasoning about the Smalltalk VM along the lines of Morisset (2017) — Armstrong et al. (2019) – Wassell et al. (2018), or more generally, deriving the VM along the lines of the BMF formalism (Bird & de Moor, 1996; Backhouse et al., 2002). At a minimum, the objects in this framework need to have an algebra of logic — or in other words, satisfy the axioms of an (elementary) topos. Thus these type objects should correspond to their most-precise intended meaning (e.g. EvenInteger is exactly the set of even integers), compose (the result of sending

*PositiveInteger* ∩ *EvenInteger*

is just another object of the same nature), have initial ($\varnothing$) and terminal ($\mathbb{Z}$) objects, etc. In such setting, the paradox with `42`'s membership does not arise because `42` is just the set of all $x \in \mathbb{Z}$ such that `x==42`.

This paper's task, then, is naturally a part of a hierarchy of tasks, at the top of which is the author's *Programm* aimed at verification and synthesis of Smalltalk VMs by formal methods.

## 2   Two naïve approaches

Before we are ready to explain the rationale for choosing the approach we propose in this paper, let's briefly consider why we failed when we tried two other, simple approaches.

### 2.1   First-order logic programming

Auler et al. (2012) synthesize a target-agnostic, tiling LLVM compiler backend from an extension to the ArchC PDL by using an ad-hoc unification procedure. We attempted to improve Auler's instruction selector for use in a Smalltalk JIT by systematically implementing Cheng et al. (1995)'s equational rewriter. This works well, even for very complex tiles, as long as the available instructions have only trivial "theory constraints", but fails at instructions such as PowerPC's `rlwinm`. In fact Auler's "CompilerInfo" for PowerPC does not attempt to correctly a model `rlwinm`'s semantics.

This is not a fault of Auler's implementation (or our attempted improvement in Smalltalk). The root cause of the problem is the nature of first-order logic programming as a free-term calculus.

Let's explain this by using a most extreme example.

Massalin's canonical example of his Superoptimizer (Massalin, 1987) automatically finds an instruction sequence implementing the `signum` function:

```
int signum(int x) {
    if (x>0) return 1;
    if (x<0) return -1;
    else     return 0;
}
```

It would seem intuitively obvious that you need at least two comparisons and two jumps, but in reality you only need three register-arithmetic instructions; on SPARC:

```
! input in %i0, output in %o1
addcc %i0 %i0 %l1
subxcc %i0 %l1 %l2
addx %l2 %i0 %o1
```

Massalin's astonishing trick works because of certain unexpected properties of the machine arithmetic of bit-vectors and flags — obscure/counterintuitive details of the behavior of carry and overflow in two's complement. These are rooted in *theory constraints*, as opposed to *free* structural properties. Logic programming

easily unifies $(3 + 4)$ with $(X + Y)$, because this is term structure; but equating $3 + 4$ with $7$ is part of the particular theory of integer arithmetic. It is not clear how to express the arithmetic properties of two's complement carry in PROLOG or its extensions, so that an AccGen-like algorithm would discover Massalin's sequence or any of the idiomatic uses of `rlwinm`.

At this point, we started looking for something that could deal with rich theories and their combinations.

## 2.2 Direct use of SMT

Shoshitaishvili et al. (2016) use an SMT solver for static analysis of native machine binaries by symbolic execution. This appeared very attractive, for its faithful modeling of actual production ISAs. However, for our purposes this approach suffers from the problem opposite to what we described in section 2.1: while state-of-the-art SMT solvers excel at dealing with combinations of theories such as Linear Arithmetic, Bit Vectors, McCarthy arrays etc., they fundamentally are *SAT checkers* and are poor at *proving* in the PROLOG sense. In particular, they are *fragile*, i.e. prone to diverge under difficult-to-predict circumstances.

In the following discussion, we shall use the textbook example of the inductive `List` datatype, the simple problem of reasoning about the *length* function,

*length :: List 'a → Integer*,

and our Smalltalk FFI to Microsoft Z3 (de Moura & Bjørner, 2008).

A List (of elements of some unspecified but fixed sort `'a`) can be thought of as an F-algebra comprising a coproduct of `Nil/0` and `Cons/2` recursively taking an `'a` and a List. This kind of inductive type declaration is directly supported by the Z3 API:

```
listType := Z3Datatype named: 'List'.
listType declare: 'cons' accessors: { 'car'->Int sort. 'cdr'->listType }.
listType declare: 'nil'.
listType := listType create.
```

Moreover, it is possible to declare the uninterpreted function *length*:

```
length := 'length' recursiveFunctionFrom: {listType} to: Int sort.
```

and assert inductive facts about its values:

```
length value: l is: (l is_nil ifTrue: [0 toInt] ifFalse: [(length value: l cdr) + 1]).
```

At this point Z3 can prove a few interesting facts about lists; for example: The length of a singleton is 1:

```
singleton := listType cons: x _: listType nil.
self assert: (
  Z3Solver isValid: (length value: singleton) === 1
)
```

Our Z3 FFI package contains a few other fun examples.

Let's try something more interesting. The function

*append :: List × List → List*

concatenates two lists by recursion:

```
append := 'append' recursiveFunctionFrom: {listType.listType} to: listType.
append valueWithArguments: {l.ll} is: (l is_nil
  ifTrue: [ll]
  ifFalse: [ | h t rest |
    h := l car.
    t := l cdr.
    rest := append value: t value: ll.
    listType cons: h _: rest ]).
```

We want to prove that the length of the "big" list is the sum of the lengths of the two "small" lists. To a human mathematician this is obvious by structural induction on the datatype, and indeed it is easy to formulate as a Z3 assertion:

```
xs := listType mkConst: 'xs'.
ys := listType mkConst: 'ys'.
self assert: (
  Z3Solver isValid:
    (length value: xs) + (length value: ys)  "|xs|+|ys|"
      ===
    (length value: (append value: xs value: ys))  "|xs,ys|"
  )
```

Unfortunately, that last Z3 API call diverges in an infinite loop. Worse yet, it is almost unrealistic to predict when an e-match will diverge; so these capabilities of Z3 are unsuitable for use in automatic proof. There is a limited fragment of the logic which is guaranteed to be decidable; we are confined to it.

In the past decade, a number of algorithms for reducing inductive proof to the decidable fragment, has been studied extensively. Of course, in general this is not possible because of undecidability; so we are only dealing with "solvable cases" in the Ackermann (1954) sense.

One of these algorithms is FUSION described in Cosman & Jhala (2017) and used in their implementation of LQDT (Rondon et al., 2008). We studied FUSION and have implemented it in Smalltalk, *mutatis mutandis* due to our use of Smalltalk as opposed to Cosman–Jhala's Haskell implementation.

These differences are of three kinds.

- The nature of interfacing with the SMT backend. Data at all stages along the transformation pipeline — the goal theorem, the Horn query, the tautology ultimately to be checked by Z3 — are live objects which can be inspected and manipulated in the debugger. This is in contrast to the batch-like SMT-LIB processing in LiquidFixpoint.

- In MachineArithmetic, the logical formulae serving as the refinement predicates are arbitrary Smalltalk code which compiles to opaque Blocks closing over transparent logical objects. In Liquid Fixpoint predicates are *transparent* AST-tree terms processed by Visitor-like folding; reflecting arbitrary Haskell code back into refinements requires the additional technique of Proof by Logical Evaluation (PLE) (Vazou et al., 2017).

- Front-end interaction of MachineArithmetic with the host Smalltalk had to be designed from first principles, due to the difference between Smalltalk's and Haskell's programming paradigm.

Figure 1 shows the principal components of MachineArithmetic. The engine is the three layers in the middle; on either side is syntactic sugar.

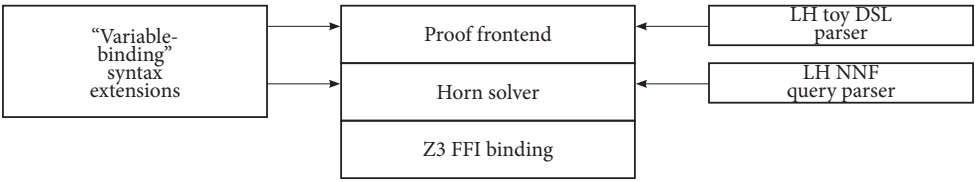

Figure 1: Principal components of MachineArithmetic.

## 3  Refinement types

In the communities of programming language design, optimizing compilers, theoretical computer science, logic, and many others, refinement types have amassed a large body of research and available literature since their introduction by Freeman & Pfenning (1991).

A refined type is built from a carrier set[3] (aka "unrefined type") and a predicate by "set comprehension". One can think of it as a subset of the carrier set: the predicate acts as the characteristic function of the subset, it is a block that takes an element of the carrier set and answers `true` if the element belongs to the subset, `false` if it's outside of the subset.

For example,

```
Integer | [ :x | x < 0 ]
```

is the set of all negative integers. The above is just normal Smalltalk; `#|` (pronounced "*such that*") is no different from any other binary selector[4].

Another simple example (let's call it $\sigma_3$ just for the next few pages) is the set of all three-character Strings:

```
String | [ :x | x size = 3 ]
```

Then $'big' \in \sigma_3$ and $'pig' \in \sigma_3$, but $'Smalltalk' \notin \sigma_3$.

One important special case of refinement is the singleton: i.e., the set $\{a\}$ given some particular/fixed element $a$. For example, the particular Integer, `42`, gives rise to the set

```
Integer | [ :x | x == 42 ].
```

This construction, which Ou et al. (2004) call "selfification", generalizes to "the most precise possible type for the term". The partial order of set inclusion (or, equivalently, implication between predicates) induces a topos of refinement types.

The author's inspiration to try using refinement types as the logical basis for his VM-verification/synthesis system, came from related work by (Armstrong et al., 2019; Wassell et al., 2018) on automated reasoning about optimizing-compiler correctness using formal specifications of ISAs and weakly consistent memory models based on the LQDT flavor of refinement types.

Why is it interesing to have refinement types in Smalltalk when there exist plenty of implementations in the ML tradition, besides serving as the foundation for the author's VM-synthesis *Programm*? To the author's best knowledge, *computing-of-proofs* has previously never been attempted in a Dynabook-like medium. By

---

[3]For the current exposition, we momentarily ignore the differentiation between sets and types despite its crucial importance; see Reynolds (1983) for more discussion. In our implementation, carriers are Sorts of the underlying SMT solver.

[4]There is nothing magical inside |, either: we just create a Refinement object holding the carrier and the block. These refinement objects get passed around, embedded in Forall quantifications, ANDed by ∧, etc. until we can no longer avoid theory-specific operations — which happens at "UNSAT candidate checking" described at the end of section 4.1.

"Dynabook-like", the author means that the cognitive activity of the human mathematician relies on a characteristic sensation of immediate contact with the mathematical construction. If a proof fails, the Smalltalk Debugger affords me to directly experience what's going on in the live computation. Diametrically opposite to this, up to date the *computing-of-proofs* has been dominated by the LCF/ML tradition, which explicitly declares the very term "*debugging*" to be derogatory: "if you *need* to debug, then you are already doing it wrong."

In the author's opinion, the roots of this contradiction in approach go back to Diophantus, who uses words such as "*four*" to refer both to the literal number 4 and to what today we would call "a variable named *four*". First he considers a certain equation containing the coefficient 4; he explains how to solve the equation in this particular case. Once the understanding of the method is built up, he next asks: "for which values of *four* does this method work?" This is both algebra (symbolic *four*) and number theory (to Diophantus, only integer solutions would seem satisfactory). Following this idea, the liberation from pointwise computing leads us towards building a Dynabook with some kind of switch between our traditional test-driven mode (Diophantus' literal 4) and the ML-like mode (variable *four*).

## 4  The algorithm

Conceptually, the algorithm we use is the Burstall–Darlington encoding (Burstall & Darlington, 1977). We refer to Cosman & Jhala (2017) for the description of details shared with their Haskell implementation; here we concentrate on the differences caused by the live nature of our Smalltalk implementation. The gist of the algorithm is as follows:

In step 1, the goal theorem is transformed into a Horn query in Negation Normal Form (NNF). This query is not first-order because it contains yet-unknown predicates (usually denoted $\kappa$).

The transformation is achieved by a repetitive, bidirectional "synthesize-check" cycle.

In step 2, the Horn query is reduced to the Burstall–Darlington representation by repeatedly eliminating unknown $\kappa$-variables using the "Fold-Unfold" fixpoint algorithm.

In step 3, we pass the resulting $\kappa$-free tautology to Z3 for checking.

In counterbalance to the "rigorous" LCF/ML-tradition-style definitions of the algorithm — of which in the literature there are plenty — we illustrate by three simple examples. The first one, in section 4.1, verifies a straight-line (i.e. no loops/recursion) program (though it does contain conditional branches). The program in section 4.2 is even simpler; we include this example for extra clarity on what we said in section 2.1 about algebraic nonfreedom and combination of theories. Section 4.3 moves on to verify a recursive program by induction on datatype structure; this is where our principal variation from Cosman–Jhala happens.

### 4.1  Example 1: A straight-line program

Let's recall our `abs` example from section 1: in particular, the question about "how to capture the positivity aspect". Consider the following program in a toy programming language widely used for teaching in the LiquidHaskell community:

```
[[ val assert : bool[b|b] => int ]]
let assert = (b) => { 0 };

[[ val abs : x:int => int[?] ]]
let abs = (x) => {
  let pos = x >= 0;
  if (pos) {
    x
  } else {
    0 - x
  }
```

```
};

[[ val main : int => int ]]
let main = (y) => {
  let ya  = abs(y);
  let ok  = 0 <= ya;
  assert(ok)
};
```

Here we see an implementation of `abs`'s body, which conditionally branches on whether the argument $x$ is negative or not. We are attempting to prove that `abs`'s return value can never be negative. That is, we want a way to express this condition, and we want to build a checker that will be able to prove that our `abs` body satisfies it, by static analysis alone.

One straightforward approach would be to explicitly specify `abs`'s postcondition:

```
[[ val abs : x:int => int[ a | a >= 0 ] ]]
```

This is easy to implement and does work, but is a burden on the programmer and doesn't scale up to real programs. Instead, in the above program we leave the explicit contract of `abs` as unknown — we refer to it as a *hole* — and synthesize it at verification time from *asserting* that whatever we got back from `abs` is nonnegative.

Remarkably, `assert` is implemented entirely within our toy language:

```
[[ val assert : bool[b|b] => int ]]
```

or perhaps it would be easier to understand if we write it, equivalently, as

```
[[ val assert : bool[b|b==true] => int ]]
```

The body of `assert` is irrelevant; only its domain ("all true Booleans") matters. The idea is that, for the call to `assert(ok)` on the last line of `main` to typecheck, we need to statically prove that ok cannot be anything other than "a true Boolean".

Parsing this program using PetitParser is trivial[5], resulting in an AST tree which we show here abridged for space's sake:

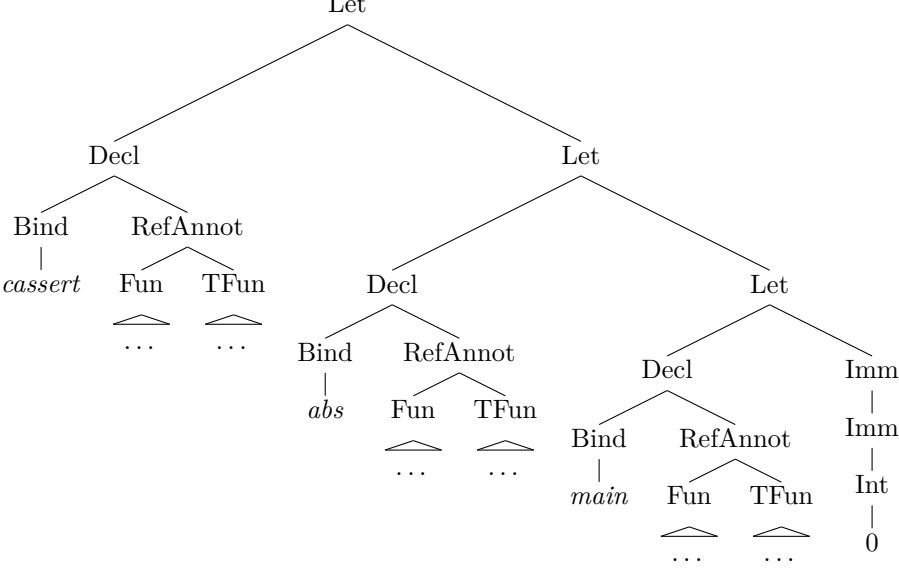

---

[5]Of course this particular syntactic frontend is just an example, and the reason it exists in MachineArithmetic is for didactic and validation purposes.

In the next phase of processing, we normalize this AST into Horn NNF form by recursive descent in a bidirectional check/synth cycle. We start with `#check:rtype:` of the root *Let* node.

`Let>>check:rtype:` is the Smalltalk implementation of the following Gentzen-style derivation rule:

$$\frac{\Gamma \vdash e \Rightarrow s \qquad \Gamma, x \in s \vdash e' \Leftarrow t'}{\Gamma \vdash let\ x = e\ in\ e' \Leftarrow t'} \text{ [Chk-Let]}$$

That is, to check the conclusion at the bottom (the *Let*), we in turn call `#synth:` on the *Decl* (the left child in the above AST diagram) and `#check:rtype:` on the premiss *Let* (the right child).

Continuing this traversal recursively, applying rules similar to the "[Chk-Let]" above at each node, ultimately produces the following Horn VC:

$$\forall x \in \mathbb{Z} \mid true.$$
$$\forall pos \in \mathbb{B} \mid pos \Leftrightarrow x \geq 0.$$
$$\& \begin{cases} \forall g & \in \mathbb{B} \mid pos. \\ & \forall v \in \mathbb{Z} \mid v = x. \quad \kappa(v, x) \\ \forall g & \in \mathbb{B} \mid \neg pos. \\ & \forall v \in \mathbb{Z} \mid v = 0 - x. \quad \kappa(v, x) \end{cases}$$

(Here the precedence of the bar "|" and the dot "." is so that "$\forall x \in S \mid A.B$" reads "$B$ holds for all $x$ in $S$ such that $A$".)

Here $\kappa$ is the Horn unknown. It corresponds to the *hole* we talked about, and captures the meaning of `abs`: $\kappa(v, x)$ is a predicate taking an integer $v$ and an integer $x$ and returning `true` if $v$ is the absolute value of $x$, `false` otherwise. But the system doesn't yet know $\kappa$; this is the work for the Horn solver, to synthesize it by solving the above NNF query. How does the Horn solver do this? If you take a formula such as $\forall x \in \mathbb{Z} \mid A.B$, $\kappa$ may occur in it in the hypothesis (the $A$ in $\forall x \in \mathbb{Z} \mid A.B$) or in the conclusion (the $B$). So these create a lower bound and an upper bound for $\kappa$. The remarkable thing is that these bounds determine $\kappa$ uniquely (intuitively speaking, there is zero wiggle room in-between). Therefore we can unambiguously substitute the arguments to $\kappa$ until all $\kappa$-unknowns are eliminated, leaving only Z3 tautologies (called "UNSAT candidates"). It is at this final stage that the predicates in refinements (the latter being built up in `#|`) become Z3 ASTs: at this point the "Alpha-uniquifier" has already renamed all potential name clashes, and we know the unrefined sort of every variable. We create leaf ASTs for each variable by calling `Z3_create_const()`, and put them into a new Smalltalk `Context`. Then the arbitrary Smalltalk blocks in the refinements constituting the UNSAT candidates, fire in that Context, returning a Z3 AST — it is that AST that we pass to Z3 to check. Now Z3 is guaranteed to not diverge because the tautologies are of the form

$$\forall x \forall y \forall z \ldots p(x, y, z, \ldots) \Rightarrow q(x, y, z, \ldots)$$

where $p$ and $q$ are purely propositional (i.e. containing only connectives but not quantifiers) formulae over atoms from decidable theories; what we are asking Z3 is whether

$$p(x, y, z, \ldots) \wedge \neg q(x, y, z, \ldots)$$

is unsatisfiable, which Z3 is guaranteed to decide efficiently.

## 4.2 Another straight-line program

Our next example is similar to the previous one. We find it illustrative because it explains our point about "theory-specific constraints". Here, we prove that the function `square` cannot ever return a negative value. For this, we are at the mercy of Z3's knowledge about the Integer Arithmetic theory. Theres is no structural information from which to deduce $x * x \geq 0$.

```
[[ val assert : bool[b|b] => int ]]
let assert = (b) => { 0 };

[[ val square : x:int => int[?] ]]
let square = (x) => {
  x * x
};

[[ val main : int => int ]]
let main = (x) => {
  let squared = square(x);
  let isPos = squared >= 0;
  assert(isPos)
};
```

This normalizes to the following Horn:

$$
\begin{cases}
\forall x \in \mathbb{Z} | true. \\
\quad \forall v \in \mathbb{Z} | v = (x * x).\kappa(v, x) \\
\forall y \in \mathbb{Z} | true. \\
\quad \forall squared \in \mathbb{Z} | \kappa(squared, y). \\
\quad\quad \forall isPos \in \mathbb{B} | isPos \Leftrightarrow squared \geq 0. \\
\quad\quad\quad \forall b \in \mathbb{B} | (b \Leftrightarrow squared \geq 0) \wedge (b = isPos).b
\end{cases}
$$

which is solved similarly to the previous section.

### 4.3 Inductive

Recall from section 2.2 the diverging attempted proof that the lengths of sub-lists add up to the length of the whole list. Here is the proof in our toy language:

```
[[ measure len : list('a) => int ]]

type list('a) =
  | Nil                      => [v| len v = 0]
  | Cons (x:'a, xs:list('a)) => [v| len v = 1 + len(xs)]
  ;

[[ val append : xs:list('a) => ys:list('a) => list('a)[v|len v = len(xs) + len(ys)] ]]
let rec append = (xs, ys) => {
  switch (xs) {
    | Nil         => ys
    | Cons(h, t) => let rest = append(t, ys);
                    Cons(h, rest)
  }
};
```

This needs surprisingly few new elements to normalize it to the same NNF form.

First, we see that **len** is a *measure*, a function that has no body and exists only as ghost code. In Z3 it instantiates to an *uninterpreted function*: nothing is known about **len** other than that it goes from some uninterpreted sort "(list 'a)" (which we shall discuss in the next section) to $\mathbb{Z}$, and that it returns equal values for equal arguments.

Normalizing the constructors "list = Nil | Cons" and deconstructors "switch = Nil | Cons" is straightforward.

Measure function application, such as `len v`, is trivial, amounting to a call to `Z3_mk_app()`.

### 4.4 The hotel of carrier sorts

A problem particular to our Smalltalk implementation, arises when a carrier type is parametrized. We saw an example of this when we wrote

*length :: List 'a → Integer.*

In general, such parametrization leads to a tree of carrier types representing the well-formedness of function-applications. In Cosman & Jhala (2017) this tree is built via unification of type terms by a pair of processes called "monomorphisation" and "elaboration", which is possible because these type terms are just ASTs existing as regular Haskell data before the final tautology is passed to Z3. In Smalltalk this is not possible because carriers must exist as live objects pointing to actual Z3 sorts in Z3's memory. While it is possible to create the uninterpred sort `'a`, this breaks when we try to verify a call to a more-general function with a more-specific actual parameter sort: such as calling a function taking a list-of-anything with a list-of-integers, because `'a` is not a "supersort" of `Integer`. In other words, such uninterpreted sort `'a` is on the wrong side of "introduction rule" — expressing "*given a fixed a, ...*" instead of "*for all a, ...*". Following Reynolds (1983), Wadler (1989) and de Bruin (1989), we can say that

*length :: List 'a → Integer*

is a natural transformation: a family of morphisms *(List Integer) → Integer*, *(List Character) → Integer*, etc. In fact, this example is in a certain sense misleading, because the family is degenerate on the target side (the codomains of every morphism are all same). It would correspond to de Bruin's construction more closely if instead of Integer, each of these functions' return type would be something different depending on the argument type (aka, the morphism's domain). This immediately suggests to us the generalization stemming from the fact that the return value's refined type is dependent on the argument value, and then that values are types, via injection by selfification (Melliès & Zeilberger (2015); Melliès & Zeilberger (2015)).

How can we represent this natural transformation in Smalltalk? The first thing to worry about, is well-formedness with respect to matching the unrefined (carrier) types, because Z3 sorts don't unify so we can't construct a lattice of most-general-unifiers. The crucial observation is that no recursion is involved in the construction of these sort trees, so the total number of nodes is always finite. We place each sort node into a "hotel of sorts", which is an instance of the class we call `IndexedSet`. This is a Smalltalk Set where each element is assigned an integer index, and this assignment does not change if the Set mutates. Imagine each node living in a numbered room in a hotel. Then we create a structureless, numbered uninterpreted Z3 sort, — unless we already unified the node with a sort we created previously, — one sort per room number. The second crucial observation is that this renaming of sorts does not change the outcome of the SMT check. This gives us our required representaion of "for all `'a`".

## 5   Some syntactic sugar

To allow natural notation for quantified expressions, Horn clauses, etc, we generalized Smalltalk's syntax for "dummy" bound variables (Shingarov, 2022).

In mathematics, "bound" (aka "dummy") variables are everywhere. Perhaps the first example of bound variables a teenager mathematician encounters, are integration variables: familiarizing with the idea that $\int_1^2 f(x)\,dx$ is the same as $\int_1^2 f(y)\,dy$, leading to the concept of $\alpha$-equivalence. Further the student meets with $\lambda$-terms (equivalence classes of preterms modulo $\alpha$), first-order quantifiers $\forall$ and $\exists$, higher-order Horn quantifier $\star$, and so on.

In Smalltalk, we write

```
[ :x | ...something... ]
```

for *lambda*-abstraction. When the compiler encounters this syntax, it emits code that at runtime will create the corresponding `BlockClosure`. This has worked well for several decades, and this author is pretty content with this design. What the author is not content with, is that this is limited to only *lambda*-abstraction and disallows all other kinds of quantification. How can we remove this limitation, without increasing the complexity of Smalltalk's simple LL(1) syntax which fits on a postcard?

The insight is that the syntactic *shape* of [ :x |...] says nothing about the *kind* of the binding – only that $x$ is a bound variable. That information about the *kind* is encapsulated inside the ":" token. So, change 1) is: we remove the knowledge about ":" hard-coded in the Scanner, and replace it with a user-defined Dictionary which maps ":", "∀" etc to classes containing *kind*-specific behavior. Change 2) is: we remove all *kind*-specific behavior out of the Parser and the backend (for example, the knowlegde how to create a `BlockClosure`) and replace it with a reference to the value in that Dictionary. In practice, it is even possible to vary local shape per kind, without escaping out of the kind-specific user-defined class. For example, we accept

$$[\forall i \in \mathbb{Z}[i < 0] \mid (A \ at\colon i) > 10]$$

which intuitively reads "for all negative whole i, (A at: i) exceeds 10".

As mentioned above, the author was careful to do this syntax extension without increasing the complexity of existing syntax. In fact, the result is added conceptual clarity, because dummy-variable capture is now separated from the block-application semantics: a $\lambda$-block such as $[\colon x \mid x * 2]$ can have "a value at $x = 5$" only because $\lambda$-abstractions have an "apply" operation attached to them, but in general simply capturing a dummy variable in a lexical unit is not enough to create such an operation; so it is meaningless to ask "what is the value of $[\forall x \mid x * 2 \geq 0]$ at $x = 5$." Our extension explicitly disentangles these two concepts.

In this notation, Smalltalk's nested lexical environments express type-dependency in an especially natural way: in

$$[\forall x \in \mathbb{Z} \mid x * x \geq a]$$

the closed-over $a$ comes from the environment immediately outside of our block. This outer environment can also be a $\forall$- or $\lambda$-block, etc.

Thus, currently in MachineArithmetic there are three ways to create objects such as *Refinement, Forall, Exists* etc.

- Explicitly by sending an instantiation message (think `#new`) to the corresponding class.

- Using the above-described syntax.

- Using the parser for LiquidHaskell DSLs.

## 6 Results and Future work

This paper describes a work in progress. Currently we are reporting the following interim empirical result:

### 6.1 Experimental results

We implemented MachineArithmetic and as the first step in validating our implementation, verified that it passes tests from the LiquidHaskell benchmark suite (which we parse using the above-described technique). We chose this approach as fitting ideally with the spirit of Smalltalk and Dynabook, which affords effortless transition between intuition-building and precise abstract definitions.

## 6.2 Proof of the algorithm

In the next step, we intend to formally show that the theorems in Cosman & Jhala (2017) continue to hold in the live case.

## 6.3 Separation logic and Optics

Most importantly, our ultimate intent is to model memory and register access (Morisset, 2017; Armstrong et al., 2019; Wassell et al., 2018) as frames (McCarthy, 1970; McCarthy & Hayes, 1969) within our formalism, thus giving an axiomatization of the GET, PUT, STORE, LOAD operations in Shoshitaishvili et al. (2016). For this ambition we are still missing a few major components. Crucially, optics is essentially coalgebraic in nature, but we have not yet implemented coalgebra. We hope to do this systematically by leveraging the Arrows system (Notarfrancesco, 2022).

## 6.4 Deep language integration

Our current codebase is hosted in a traditional Smalltalk-80 as the implementation substrate. As such, we currently have the class *Int* (for Z3 integers) distinct from Smalltalk-80's *Integer*, *Bool* (for Z3 booleans) distinct from *Boolean*, etc. One future direction of work is towards a linguistic fusion of these conceptual strata without breaking Smalltalk.

## 6.5 Use outside the original motivation

An important goal for the future is to work with others towards applications other than our original motivation (compiler verification). In Alan Kay's words, Gutenberg's press becomes interesting the moment we apply it outside its original purpose of reproducing copies of the Bible **faster**. Being an implementation of LQDT, MachineArithmetic is a general intuitionistic reasoning engine. As such, for one possible "outside context" application, it would be interesting to use MachineArithmetic as the substrate type-checker for an LCF-style engine such as a Lean kernel, thus making the body of mathematics in the Xena project available in a Dynabook-like debuggable medium.

# 7 Availability

The complete source code is available from the author's GitHub page under MIT license.

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

## Acronyms

**AST** Abstract Syntax Tree. 4

**BMF** Bird–Meertens Formalism. 2

**ISA** Instruction Set Architecture. 1

**JIT** Just-in-Time compiler. 2

**LCF/ML** Logic for Computable Functions / Meta Language. 6

**LLVM** Low Level Virtual Machine. 2

**LQDT** Logically Qualified Data Types, abbreviated to Liquid Types. 4, 9

**NNF** Negation Normal Form. 5

**PDL** Processor Description Language. 2

**PLE** Proof by Logical Evaluation. 4

**SMT** Satisfiability Modulo Theories. 3

**VC** Floyd–Hoare Verification Condition. 7

**VM** Virtual Machine. 1

