# OpenReview forum: "Live proof-by-induction"
_FAST.org.ar/2022/Workshop — FAST Smalltalk 2022_

### Official Review · Reviewer_iuDh · 2022-10-26
**A new tool available in Smalltalk to help program verification.**

**Rating:** 6
**Confidence:** 3

**Review:**

## Paper summary

This paper:

- Presents and uses a Smalltalk FFI implementation of Microsoft’s Z3. As a side note, Z3 requires citation. I suggest using the one shown below as [3].
- And more importantly, it describes a Smalltalk implementation of the FUSION algorithm for local refinement typing from Cosman and Jhala (2017) [1].

## Strengths

This paper is part of a scientific program aimed to introduce the use of formal methods to aid the development of virtual machines for Smalltalk, a program highly relevant to the community. In particular, this paper presents a novel Smalltalk implementation of an algorithm that, before this work, only existed in Haskell. Access to that algorithm directly expands the frontier of what is possible to do in this field. It puts Smalltalk researchers close to the State of the Art research on formal methods applied to VM implementations and verification.

Remarkably, the implementation of the FUSION algorithm presented in this paper is not just a Smalltalk version of the algorithm. It is also written in a way as to “have these tools homogeneous with our Smalltalk system.”
In that regard, I enjoyed the introduction of the #| binary operator to refine classes using Blocks.

## Weaknesses

The most notorious weakness of this paper is the presentation style.

At a high level, I see three different tasks being referred to in this paper:

1. The questioning about the *type* information captured by LiveTyping [Section 1]
2. A task briefly referred to, related to a Smalltalk JIT [Section 2.1].
3. The verification that the length of appending two lists equals the sum of the length of each list. [Section 2.2]

These three Tasks are different in nature, so I ask: What is the target Task on which this project goes? Following the introduction, it seems that Task 2) is the primary goal of this paper. However, it is the least described one.

Then, the author should describe the Task explicitly as the problem setup. Additionally, the presentations of Tasks 1) and 3) appear distracting. Take, for instance, Task 1). Why is SmallInteger not just right? After all, it depends on our use of typing information. Suppose, for example, that we use type information only to predict messages sent to the returned object for code completion purposes. In that case, to return SmallInteger is just right. I also need further clarification on extending the methods proposed in [1] to LiveTyping.

## Further comments

The following are section-specific comments aimed at helping the author improve the manuscript.

I think that the author should expand section 4. And this is especially true of section 4.1, which I failed to understand.

Section 4.1:

- Since this is one of the most critical parts of the paper, where the actual contribution happens, I would have appreciated more details. For instance, it would be nice to exemplify the problem with a concrete example, showing how the tree from [1] would be built. And then how that would be the case for the Smalltalk implementation. Why can "unification" not be used in the Smalltalk implementation? The answer to that question requires more elaboration. As a reader, I'm very curious about this.
- I need clarification on why $\mathrm{length}$ is a natural transformation. I would expect that to be true of functions $f\colon U[\alpha] \to V[\alpha]$, where $U[\alpha]$ and $V[\alpha]$ are type expressions functorial on $\alpha$ (in the sense of [2]). However, I don't see that in the example. Even if it is true, it is unclear how this abstraction is useful for understanding the problems we need to solve. But, in general, I think I don't understand the actual problem, and I would love to.

In section 2, starting from “One of these algorithms…", there is a list of some of the paper's contributions.

- The second item described says that in Machine Arithmetic, the logical formulae serving as refinement predicates are arbitrary Smalltalk code. It would be very interesting to me to see examples where this extra capability is of use.
- In section 2.1, the claim regarding ``rlwinm`` needs supporting evidence: either a reference or a proof.

In section 3: I'm curious about the actual semantics of the “*such that”* operator.

[1] Cosman, Benjamin, and Ranjit Jhala. "Local refinement typing." *Proceedings of the ACM on Programming Languages.* ICFP (2017): 1-27.

[2] De Bruin, Peter J. *Naturalness of polymorphism*. University of Groningen, Department of Mathematics and Computing Science, 1989.

[3] Moura, Leonardo de, and Nikolaj Bjørner. "Z3: An efficient SMT solver." *International conference on Tools and Algorithms for the Construction and Analysis of Systems*. Springer, Berlin, Heidelberg, 2008.

---

### Official Review · Reviewer_DmBj · 2022-10-26
**?**

**Rating:** 6
**Confidence:** 2

**Review:**

I have background on PL research, not much in formal methods.  Accordingly, I am not able to fully evaluate the soundness and the level of contributions of the article. I will just give my best opinion up to my knowledge hindered by the above limitations.

## Summary
I think the idea of having tools with the ability of helping us to formally reason about ST programs is not only amazing but also mandatory. Having them implemented in ST, as the smalltalk people eating our own food always advocate for, I find it cool and interesting. So, regarding the author journey I infered from the paper, just chapeau!

Now, my opinion is that the paper have limitations regarding its presentation.

The first and most important is that it does not try to be inclusive or accessible to non fully experts that have the full math background. For instance, I would expect this kind of paper to have a background section. I understand that the paper is a workshop and that there are probably #page limitations, but still I think there is a lot of non-essential content that could be skipped or deprioritize against a couple of concise and concrete bg paragraphs.

The second general feedback in terms of presentation is that the reader should make a huge effort to understand the narrative, the main contribution, and connect all the pieces. Concretely: The abs talk about an implementation of refinement types. Ok. There are a bunch of those. We want to have it in a language such as smalltalk. Ok, nice. What are the challenges there? What are the characteristics of ST that makes it interesting for having a paper about this implementation. Instead, What I read in the first couple of pages throw a bunch of keywords that are very hard to connect to the main goal: refinements type in ST. These are: VM implementations, SC memory models, Smalltalk structures inconsistencies, SMTs, Compiler synthesizers, binary translators, Live typing, etc.

## Some concrete details

Abs talks about `MachineArtithmetic`. Then at some point an algorithm is presented and it is not clear enough that that is exactly MachineArtithmetic.

We need automatic reasoning tools ok. Why use VM as motivation? Is this an author journey? Why not be specific about it? Why refinement types? If refinements types is the proper for the problems the author want to solve I found that connection missing. For instance compiler optimizations in non SC memory models. They use refinement types to prove correctness?

I found the LiveTyping example pretty disruptive. Kind of the assumption of LiveTyping is the limitation of ST for precisely giving IDE information on references and as such provides an approximation based on classical dynamic profiling techniques. How this is connect to the `no structure that would satisfy the axioms of a topos;` problem. What if I extend `SmallIntegers` with a PositiveIntegers class? What if even the tool could give me possitiveIntegers but the `ZZZ` class implements `abs`. Why not relying on type annotations "a la" typescript for instance to provide reasoning help. I am not saying I do not see the reasoning limitation, I guess my main point is "any" IDE tooling could be stated as to help reasoning about programs but I do not see LiveTyping as an example to connect with refinement types.

Then the paper jump to other tried approaches. Why? It is not clear if the limitations were of the approaches or of the instantiation of the approaches in the Smalltalk world. If it were the approaches, why mentioned them here as something tried? What is the connection of the paper with Auler's compiler synth (tiling llvm compiler, first time I head this concept). What is the relation with ref types? What are the problems with `PowerPC’s rlwinm`. How would that help me reason about abs?

Mostly the same with the second method. The method diverges to in trying to reason about a paramterizable list. Is this a method limitation?

Then the paper jumps to an algorithm. I see is a type inference by following a ref. But I would expect that to be clarified. It could have been a checker or I am not understanding anything?

Then there is an interesting statement. All Liquid Haskell tests pass. I would expect much more here. How you connect your implementation with liquid haskell tests for instance.

## Some sentences I found hard to dissect
the very approach where the systems programmer writes and tests code based on some intuitive argument why the code works, is — out of necessity — being replaced by programmatic manipulation of said argument (suitably formalized)

`have led him to realize that in Smalltalk-80 our basic constructions lack certain desirable mathematical properties.
In our arsenal of concepts, there is no structure that would satisfy the axioms of a topos; therefore we can’t build a machine logic `
Which concepts? What does it mean that our concepts do not compose? Classes?

I see this sentence after introducing live typing -> classes don’t afford us logical connectives between propositions ->
Isn't this kind of trivial for a polymorphic reflective "very" dynamic language?

`What granularity would be appropriate to capture the essence of #size?` -> Inside the liveTyping example this looks weird. Clearly not by using a dynamic approximation tool

`The root cause of this, is that first-order logic programming is fundamentally a free-term calculus, and extensions such as Prolog’s is/2 do not scale up to the arithmetic complexity of rlwinm`

They fundamentally are SAT checkers and are poor at proving in the Prolog sense -> Isn't this like well established knowledge? Like background?

`refinement types have emerged as a modular and programmer-extensible means of expressing and verifying properties of polymorphic, higher order programs in languages like ML [Dunfield 2007; Rondon et al. 2008; Xi and Pfenning 1998], Haskell [Vazou et al. 2014b], Racket [Kent et al. 2016], F ♯ [Bengtson et al. 2008], and TypeScript [Vekris et al. 2016].` `-> So why not to use the typescript approach, a language that from the same "family" as smalltalk for instance, to help introduce some concepts of your approach?

`Fusion then synthesizes the most precise refinement types for all intermediate terms that are expressible in the refinement logic, and hence checks the specified signatures.`

`In the communities of programming language design, optimizing compilers, theoretical computer science, logic, and many others` -> I did not found many articles relating ref types with compilers which actually was the main motivation. Are there many? Some citations maybe?